# *Parthenium hysterophorus*’s Endophytes: The Second Layer of Defense against Biotic and Abiotic Stresses

**DOI:** 10.3390/microorganisms10112217

**Published:** 2022-11-09

**Authors:** Asif Khan, Sajid Ali, Murtaza Khan, Muhammad Hamayun, Yong-Sun Moon

**Affiliations:** 1Laboratory of Phytochemistry, Department of Botany, University of São Paulo, São Paulo 05508-090, Brazil; 2Department of Horticulture and Life Science, Yeungnam University, Gyeongsan 712-749, Korea; 3Department of Botany, Garden Campus, Abdul Wali Khan University Mardan, Mardan 23200, Pakistan

**Keywords:** endophytes, *Parthenium*, stresses, phytochemicals, invasive weeds, crops

## Abstract

*Parthenium hysterophorus* L. is considered an obnoxious weed due to its rapid dispersal, fast multiplications, and agricultural and health hazards. In addition to its physio-molecular and phytotoxic allelochemical usage, this weed most probably uses endophytic flora as an additional line of defense to deal with stressful conditions and tolerate both biotic and abiotic stresses. The aim of this article is to report the diversity of endophytic flora (fungi and bacteria) in *P. hysterophorus* and their role in the stress mitigation (biotic and abiotic) of other important crops. Various endophytes were reported from *P. hysterophorus* and their roles in crops evaluated under biotic and abiotic stressed conditions. These endophytes have the potential to alleviate different stresses by improving crops/plants growth, development, biomass, and photosynthetic and other physiological traits. The beneficial role of the endophytes may be attributed to stress-modulating enzymes such as the antioxidants SOD, POD and APX and ACC deaminases. Additionally, the higher production of different classes of bioactive secondary metabolites, i.e., flavonoids, proline, and glutathione may also overcome tissue damage to plants under stressed conditions. Interestingly, a number of medicinally important phytochemicals such as anhydropseudo-phlegmcin-9, 10-quinone-3-amino-8-O methyl ether ‘anhydropseudophlegmacin-9, 10-quinone-3-amino-8-O methyl ether were reported from the endophytic flora of *P. hysterophorus*. Moreover, various reports revealed that fungal and bacterial endophytes of *P. hysterophorus* enhance plant growth-promoting attributes and could be added to the consortium of biofertilizers.

## 1. Introduction 

Biological invasion is one of the most important components of the human-attributed environmental change that leading to biodiversity loss and the disruption of stable ecosystem [1,2]. According to The International Herbicide-Resistant Weed Database, different global economic liabilities exceed billions of dollars to avoid/decrease the disruption of ecosystems toward the sustainability of natural ecosystems, thereby increasing food security and agriculture productivities [3,4]. More recently, understanding the frequency of invasive species in a new environment, its growth condition/rate, damage to crop quality and quantity and its possible future prospects for the betterment of native flora is a key responsibility for a country itself and neighboring countries [5,6]. *P. hysterophorus* L. is an invasive annual weed with a tape root system and erect stem that becomes woody with the passage of time. The most devastating factor is that this dispersed in a very short period of time across the globe (Figure 1) [7]. *P. hysterophorus* caused highly deleterious effects on native biodiversity, animals and human health as a result of huge economic losses [8]. 

A number of variables contribute to the invasion of *P. hysterophorus* under diverse environmental conditions, i.e., high tolerance for saline conditions, germinating at 12 to 27 °C, in heavy metal and deep roots in low humidity/moisture areas [9,10]. *P. hysterophorus* grows very quickly faster, maturing within 90–120 days, has adaptability to photo-thermal conditions and most importantly lacks natural enemies, even outside its native range [11]. As with other invasive plants, *P. hysterophorus* is a prolific seeder; an individual plant sheds approximately 150,000 seeds in its lifetime and remains viable in the soil for three years, strongly dispersed by animals and humans [12,13]. According to Weyl et al., *P. hysterophorus* has a widespread native range from Central America to South America; however, it was first investigated outside its native range in the 19th century (1810 in India and 1880 in South Africa) [14]. Mao et al. recently identified that *P. hysterophorus* has invaded 46 countries around the world [15]. Similarly, *P. hysterophorus* is supposed to have entered the Asian subcontinent into India through food grains that were imported from the USA; there, it was identified in Pune in the Maharashtra district in 1955 [16]. Later on, this species spread in the nearby countries of Pakistan, Bangladesh and Nepal via road connections [17]. 

The uniqueness of *P. hysterophorus* is based on some special biochemical and physio-morphological adaptabilities and the biosynthesis of novel phytochemicals under different conditions [18]. Moreover, *P. hysterophorus* can grow in different habitats and tolerate severe ecological conditions such as drought, salinity, high-intensity light, cold, waterlogging and heavy metal stress compare to other plants [19]. The capabilities enable *P. hysterophorus* to invade and grow in a wide range of habitats may also be a second layer of defense in the form of fungal and bacterial endophytes occurring inside plant tissues .*P. hysterophorus* colonizes by fungal and bacterial endophytes, and a number of studies have determined that *P. hysterophorus* strongly withstands different biotic and abiotic stresses due to the most competent endophytic flora [20,21]. The novelty of the current review is based on the lack of reports describing the diversity and importance of endophytes isolated from *P. hysterophorus*. Here, we report the first ever study to take such claims into consideration. The aim of this review is to document *P. hysterophorus*’s endophyte (fungi and bacteria) diversity and their role in the mitigation of biotic and abiotic stresses. Moreover, we discuss the phytochemical profile and application of the endophytes reported in an engineering plant microbiome for sustainable agriculture.

## 2. Positive Aspects of *Parthenium hysterophorus* L. 

### 2.1. Soil Improvement

Although, majority of the researchers advocate eradication of *P. hysterophorus*, however, on the other hand some researchers claiming its uses as a green manure and manure for the crops as bio-pesticides [22]. It has been found that *P. hysterophorus* green leaves compost significantly promoting grain yield and plant height in ratoon and rice crops [23]. Similarly, studies have also highlighted the capabilities of *P. hysterophorus* manure that increases carbon status of/in the soil [24]. Additionally, the leaves of *P. hysterophorus* are rich in phytochemicals like, phenolic acids, sesquiterpene, lactones like parthenin and coronopilin, alkaloids, saponins, flavonoids, tannins and unsaturated sterols that strongly inhibits other weed’s populations in the rice field, when added as a manure, replicating its bio-controlling capabilities [25]. 

### 2.2. Health Benefits

It was found that the decoction of *P. hysterophorus* L. have been used traditionally to treat a number of health issues like, urinary tract infections, fever, gastro vascular problems, malaria, diarrhea and dysentery [26]. Similarly, Maishi et al., investigated the ethnobotanical uses of *P. hysterophorus* for the treatment of eczema, pain, cold, heart issues, skin, herps and skin problems [27]. Moreover, *P. hysterophorus* has noted to be pharmacologically active as a strong analgesic in the muscular rheumatism remedies against hepatic amoebiasis. As, parthenin is a major component of *P. hysterophorus*, exhibit significant medicinal attributes like cancer, analgesic and cardiovascular diseases [28,29,30]. Similarly, the methanolic extract of flowers showed significant anti-tumor and parthenin was found that displayed significant cytotoxic potential against *T* cell leukaemia, HL-60 and Hela cancer cells lines, respectively [31]. Chandrasena et al., reported the anti-tumor potential of *P. hysterophorus* extract via; in vivo and in vitro approaches and noted significant positive results in which the reduction in size and over all tumor’s cell survival was increased [32]. In this way, Patel et al. reported the aqueous extract of *P. hysterophorus* was found to be highly hypoglycemic potential against alloxan-induced diabetic rats via and in vivo, while the extract from the flower of *P. hysterophorus* could be used for drugs development to treat the diabetes mellitus [33].

### 2.3. Role in Enzymes Production and Eradication of Other Weeds

*P. hyterophorous*’s substrate could be used for the production of important enzymes called xylanases which is a hydrolytic class enzymes that cleave xylans [34,35]. The end product of this reaction have industrial application as an artificial sweetener, biofuel and animal feed production, backing and textile industries and for the clarification of coffee and fruit juices extractions [36,37]. The *P. hysterophorus* low-cost raw materials, the interest in utilization xylanases for the ecofriendly bleaching of pulp in the pulp industries increased [38]. 

Some weeds like, *Salvinia molesta* Mitchell, *Pistia stratiotes* L. and *Eichhornia crassipes* were found devastating that causing huge disturbance in the aquatic environment there by overpowering their life cycles [39]. However, *P. hysterophorus*’ leaf dry extract significantly inhibiting the growth and development of such weeds by wilting and desiccation [40]. Similarly, it was also found that upon increasing the leaves extract’s s concentration of *P. hysterophorus* significantly inhibiting the germination of *Eragrostis curvula* [Schard. Nees] weeds [33]. 

## 3. Negative Aspects of *Parthenium hysterophorus* L. 

### 3.1. Impact on Biodiversity 

*P. hysterophorus* L. has the potential to grow well throughout the year in almost all drastic conditions, thereby disturbing the natural ecosystem (native flora) [36]. The absence of effective natural enemy and its allopathic potential as well as other photo and thermos insensitivities is a serious tread for the natural biodiversity [41]. Due to the rapid germination, growth and shedding of numerous seeds to the soil *P. hysterophorus* can actively disturb the natural ecosystem through allelopathic suppressive interaction by releasing its phytochemicals (caffeic acid, parthenin, coronopilin and p-coumaric acid) [42]. Similarly, *P. hysterophorus* infestation to an area where the richness and evenness of the native flora’s biodiversity significantly affected [43]. *P. hysterophorus* not only disturb the crop but also causes sever diseases in animals and human [44]. It has also been taken under consideration that *P. hysterophorus* had adverse impact on the legumes by disturbing their symbiosis brought by nitrogen bacteria and fungi [45]. Besides, it also produces a very large number of pollens (700 million/plant) which are lighter and could be travel to far away distance from the source plant to other crops and inhibiting the fruiting management of such crop target plants including, beans, tomato, wheat, brinjal and other cereals [46]. It was found that *P. hysterophorus* causes drastic loss (40%) only in legumes crops, across the globe [36]. 

### 3.2. Impact on Soil Micro-Flora 

It has been taken under consideration that *P. hysterophorus* L. inhibited the activities of various bacterial and fungal species that works as nitrogen (N) and phosphorous (P) assimilations [47,48]. Studies have showed that the leaves aqueous extract of *P. hysterophorus* has detrimental effects on the biology of *Rhizobium*, *Azotobacter* and *Nitrosomonas*, respectively. It causes reduction in the legheamoglobin contents of the root nodules through which Rhizobium-legumes symbiosis is mainly affected [49]. Similarly, root and leaves leachates and their chemical components are also inhabited the nitrate production [50,51]. Additionally, *P. hysterophorus* could also negatively affect the growth and development of algae and mycorrhizal associated crops due its anti-fungal potential [52]. 

### 3.3. Impact on Live Stock

*P. hysterophorus* is a noxious weed causes serious problems in livestock includes, skin pigmentation, mouth infection, ulcer, stomach problems and diarrhea [53]. The problem severity depends on the concentration of *P. hysterophorus* and ages of plant and animals and sometimes even causing death [54]. It has also been taken under consideration that *P. hysterophorus* extract reduces the Red Blood Cells (RBC) contents in the animals which results in the weakening of immune system [55]. 

## 4. Phytochemicals of *Parthenium hysterophorus* L. 

Phytochemical investigation of *P. hysterophorus* confirmed that all its parts, even including pollens and trichmoes found richer in bioactive compounds known as sesquiterpenes lcatones (SQL), especially glycosides parthenin [56]. Similarly, other phytotoxic chemicals are ambrosin, hysterin, flavonoids, like, quercelagetin 3, 7-dimethylether, 6-hydroxyl kaempferol 3-0 arabinoglucoside and fumaric acid. Additionally, P-hydroxy benzoin, vanillic acid, caffeic acid, p-courmaric, anisic acid, p-anisic acid, chlorogenic acid, ferulic acid and sitosterol were also taken under consideration [33]. Among other parts, leaves of *P. hysterophorus* were found rich that contain a number of phenolic compounds like, caffeic, p-coumaric, gallic, p-hydroxybenzoic, anisic, vanillic and ferulic acids. *P. hysterophorus* leaves, on the other hand, contain a number of phenolic acids, sesquiterpene lactones like parthenin and coronopilin, alkaloids, saponins, flavonoids, tannins and unsaturated sterols [22,57]. 

These phytochemicals were found to be the culprits behind the threatening characters of this weed in provoking serious health hazards [58]. Interestingly, *P. hysterophorus* from diverse geographical regions showed the same phytochemicals as a major constituents (Figure 2). Das et al., investigated the flowers extract of *P. hysterophorus* and isolated four acetylated pseudoguaiamolides along with a number of known constituents [31]. Venkataiah et al., isolated a novel sesquiterpenoid, charminarone and the very seco-pseudoguianolide and other related compounds in fractions [59]. 

## 5. Endophytic Flora and Their Diverse Activities, under Stressful Conditions

Endophytes are the class of symbiotic microbes, especially the bacteria and fungi actively participating in the maintenance in the normal physio-chemical traits of plants (Figure 3) [60,61,62,63]. These endophytes were noted to produces a number of hormones, enzymes, phytochemicals and iron carriers that directly and indirectly improving the growth and development of plants, under stress conditions [64,65,66]. The endophytes actively promotes the dissolution of some essential metals and minerals there by acting as a bio fertilizer or/and producing antioxidant enzymes and hormones that increases the oxidative capabilities of host plants towards adverse conditions [67,68]. Moreover, due to the prompt urbanization, intensive agriculture-horticulture and industrialization are some key causes of heavy meatal (HM) pollutions, worldwide [69]. Besides, other pollutions/contaminants HM are non-degradable and deprived of being intervention and persist actively in the soil for several decades [70]. Study of Qadir et al. have shown that these HM significantly affects/disturb the natural ecosystem, crops quality and quantity, water quality, soil microorganisms and human health as well [71]. Concerning the physiochemical remediation methods are generally expensive, time consuming and non-friendly for the native flora, microbes and human health, however, endophytic flora of plants have the potential to convert these HMs in to least hazardous form and make it possible for the plants usage [72]. 

The key factors that causing or/and initiating the oxidative stresses are mainly related to the induction of phenylpropanoid metabolism in the plants during abnormal photosynthesis (because of the stresses) that results in the production of phenolic compounds [73,74]. However, ROS (reactive oxygen species) starting to begin in the chloroplast flowed by mitochondria and later on these ROS are penetrating out from such organelles to the rest of the cell that resulting in mutation and cell death [75]. The continuous production of such ROS in a cell/tissues because of continuous stresses like drought, heavy metals, slat and other biotic stresses plants does not survive anymore death of plant occurs. However, the presence of endophytes in the plants tissue strengthen the immune system of plants thereby producing their won antioxidant enzymes or/and stimulating the host potential too [76].

## 6. Endophytic Flora of *Parthenium hysterophorus* L.

In general, weeds are consider to be the unwanted plants that reduces the crop productivities and harm to animals and human, causing skin complication and allergies [77]. However, these plants (weeds) have also been one the problem for the natural biodiversity and environment because their competitive nature for the physical (water, space, nutrients and soil) and chemicals resources (Phosphorus Nitrogen and carbon) with the other plant and crops and ultimately hazardous to animals and humans [78,79]. Among such weeds *P. hysterophorus* L. (Asteraceae) is one of the most notorious weed around the word. Different controlling strategies (biological, chemicals, cultural and mechanical) were taken under consideration, last few decades [80]. To some extent these strategies works at very small scale, however at large scale these are not good and effective, as needed [81]. The reason could be the adoptability of *P. hysterophorus via*; molecular, phytochemical, anatomical, genetic level and phytochemicals level [36]. Besides, there is also increasing the trend for the alternate strategies, i.e., microbial communities (rhizosphere, endophytes and epiphytes) that directly linked that increasing the plants capabilities towards such extreme situations [36,82,83]. Among all, endophytes (fungi and bacteria) played a dramatic physiological role in the growth and development of host plants by producing or inducing the production of phytotoxic secondary metabolites that have active potential against various biotic and abiotic stresses [84]. These endophytes are involved directly or indirectly in the adaptation to new habitat and defensive mutualism of the host plant, i.e., *P. hysterophorus* [85].

It is generally been accepted that oxidative stresses in plants is mainly the direct and indirect consequences of different biotic and abiotic stresses that affecting the physiology of plants [71,86]. Along with plants own phenolic contents endophytic fungi and bacteria associated with *P. hysterophorus* are known to produce different phenolic and other antioxidant compounds [87,88]. Interestingly, endophytes are known to produces various classes phytohomonees, including, indol-3-acetic acid (IAA), gibberellic acid (GA), cytokinins (CK), abscisic acid (ABA), jasmonic acid (JA), strigolactnes (St), ethylene (ET) and brassinosteroides (BR) (Figure 4). These phytohomones produced by endophytes, under adverse conditions favorably overcome its negative impact over the plants normal growth and development [68,69]. 

Interestingly, more recently these endophytes were found to have the abilities to enhance the growth and development of different crops. Various growth traits for example; germination rate, fresh biomass, chlorophyll contents and desolation of essential metals (Fe, Ca. Mg, P, S and K) in the soil, improved by endophytes, under stressed conditions (Figure 5). On the same way endophytes have the potential to remove heavy metals (Cd, Mo, Zn, Cu, Hg, Mn, Br and Mg) completely or make it reasonable for the usage of crops plants, especially in contaminant soils [65,89]. Detoxification signal transduction pathways (DSTP) and metals chelation system are somehow possible mechanism through which endophytes could possibly detoxify the metals adverse impacts on plants and thereby improving the physiological and state of the plants/crops [90]. On the same way, under stressed conditions plants/crops exudates coordinates with metals and thereby increasing the metal internalization potential in plants [67]. 

A number of studies were taken under consideration to document endophytes and check its secondary metabolites production and their potential toward different stressed conditions. In this review a total of 74 endophytes were 17 were documented for different biotic and abiotic stresses (Table 1).

## 7. Endophytes from *P. hysterophorus* and Their Role Crops under Abiotic Stressful Conditions

### 7.1. Heavy Metal Stress

Heavy metals (HMS) are considered to be the natural/basic constituents of their earth’s crust, however, due to the rise in anthropogenic activities their biochemical and geochemical cycle have been disturbed [92,98]. HMs involve, Cadmium (Cd), Mercury (Hg), Chromium (Cr), Zinc (Zn), Cobalt (Co) Lead (Pb) and Arsenic (Ar) are among the leading metals that have been increasing the ecological and public health issues related to environmental contamination [76]. Some of the key factors now a days responsible for the over production of HMs are abrupt usage of fertilizer and chemicals in the fields, industries, technological and industrial applications, that increases the exposure of human beings and to HMs [99]. Similarly, plants growing in such contaminated soils are of sever risk due to the bio-availabilities of these HMs in high concentrations [100]. Studies have shown that HMs significantly affects cellular components like lysosomes, cell membrane, chlorophyll contents, mitochondria, nuclei and other essential enzymes responsible for the detoxification and damage repair system. Besides, these HMs also actively interacting with the genetic makeup of the cell by disturbing its DNA sequences, nuclear proteins and most importantly the conformational changes that causes cell cycle modulation and ultimately the carcinogenic cell [101,102]. 

Zahoor et al. isolated endophytic fungi from the *P. hysterophorus* plant growing in the HMs contaminant soil and investigate its tolerating capabilities towards Copper (Cu2), Manganese (Mn), Zinc (Zn^2+^), Chromium (Cr^6+^) and Cobalt (Co^2+^) [92]. A total of 27 strains were isolated, among all, only *Mucor* sp. MHR-7 was tolerant towards the HMs. It was found that MHR-7 strain can remove approximately 60–87% of the HMs from the broth culture when applied 300 µg mL^−1^ of such metals. It was also predicted that strain removed metals through biotransformation or/and accumulation of the HMs in its hyphae and making it less available to the plant roots, hence reducing the uptake and its toxicity in the mustard seedlings. Additionally, *Mucor* sp. MHR-7 was able to produce ACC deaminase, IAA and actively solubilizing inorganic phosphate making it’s an excellent phytostimulant fungus. It was concluded that *P. hysterophorus* growing in such contaminant soil condition may uses fungal endophytes like MHR-7 as a bio-fertilizers candidate toward such heavy metals. 

On the same way, Oves et al. isolated *Ensifer adhaereni* strain from the *Cicer arietinum* and check its heavy metals remediation like, Cr, Pb, Cd Ni and Zn [103]. It was taken under consideration that *E. adhaereni* can tolerate Cd upto 250 μg/mL, followed by 500 μg/mL. 800 μg/mL for the Cu and Zn and 1000 μg/mL for the Ni (approximately 90%), while least absorption was found in case of Pb (70%). It was also noted that these endophytes significantly tolerate the heavy metals but also solubilizes inorganic phosphate (P), secret IAA, siderophore, ammonia and HCN, under in vitro conditions. 

Kang et al. invested the copper (Cu) bioremediation and resistance mechanism potential of *Acintobacter calcoaceticus* strain KW3 being isolated from the sludge of Cu mine [104]. It was taken under consideration that effect of Cu concentrations on the bacterial biomass, adoption and growth and can tolerate Cu 400 mg/L with maximum adsorption capacity was 14.1 mg/g dry cell mass. Similarly, cell walls and intracellular and other soluble components adsorbed at 51.2 and 46.6% of Cu^2+^, respectively. The current study highlighted the ability of *A. calcoaceticus* remediation not only absorb the Cu on the surface but also metastasized ions into the cells. On the same way, Mukhtar et al. isolated and identified several members of the aspergillus genus, including *Aspergillus niger*, *Aspergillus parasiticus* and *Aspergillus reperi* [105]. However, Shang et al. reported *Aspergillus niger* as a significant bio-remediating fungal strain of lead (Pb) and phosphorus (P) from the pyromorphite source and the biomass interestingly indicated the presence of organic acids conform the dissolution of such minerals, accordingly [106]. *Penicillium funiculosum* Thorn. and *Alternaria* isolated from the *P. hysterophorus* for its pathogenic abilities, on the same way Shi et al. isolated the same endophytes from the *Brassica napus* and monitored its effect on the phytoremediation efficacy of the plant growing in the heavy metals contaminant areas. It was found that *Alternaria* sp. CBSF68 showed resistance to 1 mMCd and 10 mm Pb, respectively. While, *Penicillium* sp. CBRF65 and showed tolerance up to 1 mM Cd and 10 mM Pb and could actively produces siderosphore and IAA (indol-3-actic acid) [107]. 

### 7.2. Heat Tolerance Potential 

Due to climatic changes the average global temperature is un-expectedly increasing, however, heat stress is one of key limiting factors severely affecting future agriculture prospects [108]. Studies, have find an alternate solution to isolate and identify plant-associated microbes, especially from weeds or heat tolerated species that can promote heat tolerance in crops plants [109,110,111]. On that context, Dubey et al. screen a total of 21 bacterial strains from the *P. hysterophorus* seedlings and checked it’s for the heat tolerance potential. Among all strains, only one isolate Ph-04 was predicted to confer enhances the heat tolerance in the wheat seedlings [93]. Based on 16 s rRNA gene sequence analysis Ph-04 isolate share the sequence similarity with the *Bacillus paramycoides*. It was taken under consideration that Ph-04 treated wheat seeds promoted germination, root and shoot length, longer coleoptile, as compare to control, under dark at normal and high temperatures. Interestingly, under autotrophic conditions, as compared to control a quite significant increase in the membrane integrity and very low levels of H_2_O_2_ was found in the Ph-04 treated wheat seedlings. Additionally, this heat tolerance ability was correlated to the higher level of proline and other antioxidant enzymes, especially in the Ph-04 treated plants under heat stress and the production get increase with increase in the heat level, as compare to control seedlings. It was found that after stress the recovery of Ph-04 treated of wheat’s seedlings in terms of physio-morphological characteristics was much persistent, as compared to control. It was concluded that invasive weeds, i.e., *P. hysterophorus* have the ability to harbor potentially important endophytic microbes that can be easily transferred to non-native crops (host) plants to improve climatic resilience characteristics, accordingly. 

Rana et al. (2021) reviewed *Curvularia clavata* from the *P. hysterophorus* plant [112]. Interestingly Ali et al. isolated the same strain, i.e., *Curvularia proturberata* isolate Cp4666D from the *Triticum aestivum* and checked its heat tolerance potential [68]. Field experiment was taken under consideration and found that *C. proturberata* causing a remarkable thermotolerance in the host plants and interestingly increasing the production of Cytokinins, indo-3-acetic acid, chlorophyll contents and molecular proteins. On the same way, this train significantly suppressing the plant associated pathogens and the rate of free radicles production.

### 7.3. Role in Salinity Stress Mitigation

The average production of the crops decreasing day by day due a biotic and abiotic stresses and the scientist alarming this issue to be increase very rapidly [68,113]. Among all stresses, salinity is one of the most important abiotic stress that severely affect the arid and semi-arid areas of the world, where the insufficient rain fall could not leach salt from the root zone due to which the crop productivities loss [114,115]. Soil salinization leads to the alteration or even disruption of the natural biological [116], biochemical [117], hydrological [118] and erosional (Wang et al. 2021) earth cycles [119]. Soil salinity interferes with biological uptake of nutrients and water, thus disturbing important physiological process required for the growth and development of plants [120]. On the same way, high salinization levels could thus result to lose the emerging resources, goods and other essential services of normal healthy soil, impacting the agricultural productions (both quality and quantity) and environmental health, accordingly [121]. The effect of salinity not only restricted to plants but could also evolving into sociocultural, domestic animals and human health concerned that hinders economic and general welfare [122]. 

Similarly, soil salinity is generally, excess accretion of salt in the form of ions, including, Boron (Br^2+^), Calcium (Ca^2+^), Sulfates (SO_4_^−2^), Magnesium (Mg^2+^), Fluorides (F^+^) and Chlorides (Cl^−^) that directly interfere in the chemo-physiological and enzymatic traits of the plants due to which the growth and development severely affected [123]. Soil salinity might be alkaline and sodic in nature that are mainly characterized by the presence of high Boron (Br^2+^), Sodium (Na^+^) and Calcium cations (Ca^2+^) [124]. As, soil salinity mainly affected the offer potion of the soil causes electrical conductivity with the rate of 4 dS/m creating an osmotic pressure of 0.2 MPa and it was estimated that around 800 million hectors area were affected around the globe [125]. It was taken under consideration that the humid area’s soil are very common, especially in the arid and semi-arid region of the world [126]. However, this might not be true always because salt also affected the tropical region of Latin America, Central Asia and Africa, but salinity manly depends on the anthropogenic activities of an area, climatic conditions and distance from the sea, respectively [127]. 

However, plants are in close association with different classes of microorganism that make plants to grow and survive in such extreme saline conditions [68]. Among all, endophytes (fungi and bacteria) have the ability to produce plants growth regulating hormones (such as, ethylene (Et), abscisic acid (ABA), gibberellins (GB), brassinosteroids (BR), jasmonic acids (JA), indole-3-acetic acid (IAA) and cytokinin (CK)) and also some sort of secondary metabolites like phenols, terpenes and alkaloids, in order to help the plants against during stresses especially salt stress [128]. Among all, IAA is one of the best known growth stimulating factor, which regulate roots growth, delay H_2_O_2_ production, inhibiting ethylene-based abscission, and stimulates plants growth and development under salinity stresses ([129]. Similarly, flavonoids are also known to controlling stomatal opening and a best resources allocation during extreme stress (Peer and Murphy, 2007). Besides, several studies have established that, IAA, phenols, DPPH activity, flavonoids and alkaloids are the basic elements used by endophytic fungi to interact and able the host plants to succeed in salt stress environments [130,131].

*Parthenium hysterophorus* L. have been investigated to be one of the salt tolerating invasive weeds that can grow even in the salty soil [132]. Exposing *P. hysterophorus* to salt stress appear to form aspartate as one of the primary product of prosthesis. It was found that during salt stress the activation of PEP carboxylase also take place. Similarly, the excessive accumulation of malate during the steady state of the photosynthesis was possibly might be due to the inhibition of malic enzyme and produces aspartate producer, during salt stress. Being as weed, *P. hysterophorus* is among those plants appears to be highly adoptive toward saline condition [133]. 

Besides the plant mechanism there are several reports emphasizing the role of endophytes that withstand during salt stress [134,135,136]. Several endophytic fungi and bacteria were recently isolated from my *P. hysterophorus* that might contribute in the mitigation of salt stress. For example, Ahmad et al., *Alternaria* sp. from *P. hysterophorus*, and Azad and Kaminskyj, inoculated into the tomato seedlings under salt stress resulted in the maintenance of photosynthetic efficacy and effectively reduces the reactive oxygen species (ROS), as compared to salt stressed and control seedlings [91,137]. On the same way, *Acinetobacter calcoaceticus* isolated from *P. hysterophorus* by Mukhtar et al. and Khan et al. also isolated the same strain from the *Populus* sp. and monitored its salinity potential and found very effective against saline condition and interestingly reduces the ROS production and increment of IAA production [20,138]. Mukhtar et al. (2010), investigated the *Phoma* sp. strain from the *P. hysterophorus*, while, Kouadria et al. (2018), inoculation enhanced the salt tolerance potential against saline conditions [20].

### 7.4. Hormonal Regulation and Its Production under Abiotic Stresses 

Plants are colonized by endophytic bacteria and fungi that engineer its capabilities towards the newly/adverse habitat by producing a number of secondary metabolites with diverse and novel molecular architectures that interestingly exhibiting of biological activities [139,140,141]. The endophytes actively contributed the plant’s resistance by up-regulating a number of defense related genes in the form of Abscisic acid (ABA), IAA (Indol-3 acetic acid), Brassinosteroid (BR), Auxin, Cytokinin (CK), Gibberellic acid (GA), Ethylene (Et) and salicylic acid (SA) [68,142]. Additionally, these are also stimulatingly increases the ecological fitness of their host plants by improving, the plant growth, yield, their nutrients uptake and tolerance to biotic and abiotic stresses [143]. Additionally, these endophytes have the capabilities to lower the ethylene production with the help of an enzyme called 1-aminocyclopropane-1-carboxylate (ACC) deaminase and solubilizing of essential metals (iron, zinc, sulfur, phosphorus and potassium etc.) and hence increases the growth and survival of the plants under various stressed conditions [144]. Besides, endophytes also produces organic acids, antimicrobial compounds including, antibiotics and cyanides, essential enzymes, induces systemic resistance and siderophores that indirectly improving plant growth and developments [145]. On the same way, ACC deaminase is of key characteristics of endophytes that stimulating growth under sever heavy metals stress [146]. 

One of the interesting interaction among endophytes and rhizosphere microbes including, root-fungi and bacterial, fungi-bacterial interaction, bacterial-bacterial, fungi-fungi interactions, have the capabilities to fix nitrogen in the soil and major groups of fungi and bacteria found that inhibiting the nodules of plants [133,147]. It taken under consideration that α and β-rhizobial strains that are also found in the root of *P. hysterophorus*. This class of endophytes is key important that indirectly stimulating plant growth by producing phytohornmones against a number of phyto-pathogens [148,149]. 

### 7.5. Phosphate *Solubilization and Uptake*

Soil is considered to be the main reservoir of essential nutrients needed for plants growth and development [150]. These including, nitrogen (N), Potassium (K) and phosphorus (P) are considered to be important and collectively called NPK. However, among all P^+^ is one of the key micronutrient involve in transformation of starch and sugars, photosynthesis, genetic characteristics from one generation to the next, energy transformation and nutrients movement across the organ of plants and can easily be found in every organ and cells of living cells [151,152]. The availability of P^+^ is normally very low and not easily available for the plants, however, in the tropical and sub-tropical region of the world, due to its complex form (Al_3_^+^, Fe_3_^+^ and Ca_2_^+^) [153]. One of the alternate source to this problem is the use of synthetic fertilizers mandatory for such soils to normalized the crop productivities, but unfortunately the these synthetic fertilizers causes some hazardous effects on the biotic and abiotic components of the ecosystem [154]. Interestingly, the microorganisms especially endophytes has the capability of phosphate solubilization from its complex to simple form for the usage and make it available for plants [155]. 

Priyadharsini et al. isolated an endophytic fungi called *Curvularia geniculata* from *P. hysterophorus* and checked its phosphate (P) solubilzation *via*; in vitro approach. It was taken under consideration that *C. geniculata* have the ability to solubilize P from different sources for example [AlPO_4_, Ca_3_(PO_4_)2 and FePO_4_], while producing indole-3-acetic acid (IAA) [21]. As compared to control *C. geniculata* improving the growth and development of model *Cajanus cajan* L. plants, under different P stressed conditions. It was found that biomass, phosphate solubilization and pH of the culture medium was significantly affected by P^+^ source and time of incubation, however, interestingly the production of IAA was conformed through chromatographic analysis and its production was increased with increasing tryptophan’s concentration in a non-linear way. It was concluded that *C. geniculata* reside inside the *P. hysterophorus* tissues which mediates its growth and development by phosphate solubilization and phytohormones production. The presence of such fungal endophytes which act as a bio-inoculant, might be *P. hysterophorus* uses it as a growth regulator tool in harsh and P. deficient soil condition. 

## 8. Endophytes from *Parthenium hysterophorus* L. and Their Role in Crops under Biotic Stress

### 8.1. Herbicidal Potential of P. hysterophorus Endophytes

Weeds are considered to be economically important pests that causes loss in the agriculture and forestry sectors due to which a number of ecological issues and most of which even capable of altering the natural ecosystem displacing the fauna and native flora [156,157]. Research have been proved that plant pathogenic microorganism have also been a lucrative and novel source of bioactive compounds with some sort/degree of herbicidal prospects [158]. Among all, *Phoma* sp. is well known phytopathgen causing a number of diseases in plant clades and also produces array of bioactive extracellular phytotoxic compounds (e.g., anthraquinones) with a potential biological activities [159]. Quereshi et al. isolated a fungal strain (*Phoma herbarum* FGCC#54) from *P. hysterophorus* and check its crude aqueous and different fractions for the presence of bioactive compounds [94]. Interestingly, they identified anhydropseudo-phlegmcin-9, 10-quinone-3-amino-8-O methyl ether *via*; ^1^HNMR ‘anhydropseudophlegmacin-9,10-quinone-3-amino-8-Omethyl ether’ pigment and checked its herbicidal potential against four prominent Indian weeds namely *Lantana camara*, *Hyptis suavelens*, *P. hysterophorus* and *Sida Acuta*. It was taken under consideration that maximum significant phytotoxic effect was observed for the *P. hysterophorus* followed by *L. camara*, while the least effect was found for the *Hyptis* and *S. acuta*. It was recommended that *P. herbarum* is an important endophytic fungi reside inside the *P. hysterophorus*, releasing precious compounds that could be used as potential herbicides against many invasive weeds. 

### 8.2. Weedicidial Potential of Parthenium hysterophorus’s Endophytes

Weeds are undesirable plants species that significantly reduces the quantitative and qualitative terms of crop productivities, but also causes several serious diseases in the human and animals as well [160]. Additionally these weeds have also been one of the most serious issue for the agriculture and environment due to its competitive nature for the light, water and nutrients with other agricultural crops and forest plants as well as the hazard they possess to the animal as well [161,162]. Since from several decades the scientist are trying to control these weeds by applying chemical, physical and biological tools but doesn’t sound better [80]. However, Endophytes are the class of microorganism that living inside the plants tissues and producing a number of phytochemicals that can used as herbicidal candidates [163]. Among other weeds, *P. hysterophorus* is one of the aggressive and fast growing weeds in the approximately in the whole world and its endophytes made it superior over the other weeds. Ahmad et al. isolated three endophytic fungal strains namely, *Alternaria*, *Aspergillus* and *Drechslera sp*. from the root of *P. hysterophorus* and checked their culture filtrate against the growth and development of three other weed species *Chenopodium album*, *Convolvulus arvensis* and *Avena fatua* [91]. It was taken under consideration that among all endophytic strains the cultural filtrate of *Alternaria* sp. possess the most significant phytotoxic effect, followed by *Drechslera* sp. and *Aspergillus* sp. against the selected target species. It was also found that these endophytic fungal strains also affected the chemo-physiological traits of the weeds species. Finally, it was concluded the sue of the *P. hysterophorus*’s endophytes could be effective, low-costly, non-chemical and ecofriendly biological control approach that can utilized for the bio-management of fast-spreading and aggressive weeds. 

### 8.3. Antimicrobial Potential of P. hysterophorus Endophytes 

The development of resistance towards the available antibiotics by the pathogenic fungi and bacteria is one the key challenge globally [164]. A number of factors are accountable for the antibiotic resistance for example late diagnosis of infection, misuses, poor hygienic conditions, inappropriate use of the antibiotics and increased in the number of immune-compromised patients [84,165]. Avoiding these, scientist are trying to search new and more effective candidates for the treatment of human’s health issues, which have been noticed to increased [166]. It has been taken under consideration that many natural compounds being isolated from medicinal plants algae, fungi and bacteria are utilized for the treatment of different human being’s aliments even on clinical trials [167]. As compared to synthetic derived drugs natural compounds are preferably utilized as rich source of pharmaceutical drugs because of their biological friendly in nature [168,169]. One of the most important aspect of natural product is multi-drug resistant infectious agents whether alone or in combination with antibiotics [63]. However, the usage of antimicrobial agents of natural origin is much valuable, due to their capabilities to via; protein-protein interaction as a result microbes very rarely developing resistance toward them [84]. 

Among all, microorganism like endophytic bacteria and fungi colonizes inside the plant tissues without causing any damaging effect and being considered to be natural reservoirs of different precious bioactive compounds [170,171]. Secondary metabolites like, phenols, alkaloids, terpenes, steroids, lactones, saponins and isocoumains have been investigated from endophytic bacteria and fungi [172]. However, studies of Liang et al. & Rukaia et al. have confirmed that endophytic bacteria could also produces a number metabolites as same as or with more pronounced biological activities than that of their respective hosts [173,174]. 

Besides, other plants *P. hysterophorus* is one of the notorious invasive weeds with potential application in different fields [175]. However, this weed have the ability resist any sort of biotic and abiotic stresses and overcome in different nonnative environments/areas and disturb the native flora and biodiversity in a very short duration of time [9]. Recently, a number of endophytes including bacteria and fungi have been enlisted from the *P. hysterophorus* that produces a number phytochemicals with key pharmacological potential (Table 2). Tanvir et al. isolated a total of 42 actinomycetes from the *P. hysterophorus* roots and monitored different biological activities like antimicrobial, antioxidant and cytotoxic [95]. It was found that strains like, RT46, 49, 54, and 63 displayed foremost MIC (Minimal Inhibitory Concentration) that ranging from 4–32 lg/mL against all nosocomial pathogens, including *X4 Psedomonas* sp., *E4 Enterbacter* sp., *S2 Enterobacter* sp. and *M9 Enterobacter* sp. Additionally, it was also noted that RT6, 13, 18, 36, 50, 53, 56, 57, 59 and 60 showed significant growth inhibitory effect, while TR46 and 49 showed least effect against pathogenic microbes (fungi, bacteria and algae). Similarly, the exposure of brine shrimp to the showed an interesting output for example, RT13, 50, 59 and 67 the percent mortality after 24 h ranged from 96% at 100 lg/mL to 100% at 500, 1 and 5 mg/mL, isolates, RT18, 36 and 53 gives 100% mortality at 100 lg/mL at the lowest concentration of extracts tested. It was suggested to that these *Actinomycetes* isolated from *P. hysterophorus* possess significant biological activities which prove the presence of precious metabolites, which need a prompt response toward the isolation and identification of such compounds. Similarly, Bascom-Slack et al. also isolated the same class of endophytic *Actinomycetes* and extracted/discovered the coronamycin [176]. 

### 8.4. Parthenium hysterophorus’s Endophytes as a Diuron Degradation 

The occurrence of herbicides in the water bodies is a major concern, globally [177]. It was found that, these herbicides enter to the aquatic ecosystem through, accidental spills, leaching and runoff. Among all, *Phrnylurea* herbicides especially the diuron [*N*-(3, 4-dichlorophenyl)-*N*, *N*-dimethyl urea] and its degraded intermediate namely, 3, 4-dichloroaniline (3, 4-DCA) are listed as diuron with priority hazardous material especially for the European fresh water resources (Directive 2000/60/EC) [178]. 

Depending on the concentration (range may be vary from ng/L to mg/L) of the diuron in polluted water bodies, however as the concentration increases the interaction among the species weekend due to which loss in the biodiversity occurs [179]. It has been taken under concentration that diuron directly affect the primary producers of the aquatic food web by altering their photosynthetic and physio-chemical prospects, due to which this in turn have a detrimental outcome on the organism present at higher tropic level of food web. However, diuron have also been reported to affect the microbial communities [180]. Besides, for the elimination of such hazardous diurons from the water bodies an environmental friendly approach for the maintenance of natural aquatic ecosystem is needed. The degradation of such pollutants by manipulating the physiological potential of microorganism is an environmental friendly approach for the elimination of such pollution [181]. Remarkably, most of the microbes are utilized for this purposes have been isolated from the rhizosphere, water bodies and endophytic in origin especially from those plants surviving in such contaminant soils or areas, respectively [140]. Research on the role of endophytes in the mitigation of remediation and pollutants [182,183,184]. 

It’s believed that these endophytes residing in the internal tissues of their host plants may have a role/ helping the plants growing in contaminates soils/areas [185]. These endophytes remain in close contact with their host and significantly influenced by the chemo-physiological status of the host plants, for example plants growing in the contaminant sites have the chances of harboring pollutant –degrading endophytes [186]. These pollutants after absorption *via;* roots and roots hair are mainly transported to other parts of the plants through the vascular system and the endophytes present in the plants directly and indirectly exposed to such pollutants toxicity [187]. Endophytes evolve a suitable mechanism to mitigate/degrade the toxic effect to benign form. However, the existence of such pollutants-degrading pollutants endophytes (fungi and bacteria) in the close vicinity of could possibly empower the plants ability towards the biological detoxification resources needed for the survival in the contaminant soils [188,189].

Singh and Singla, isolated plants growth promoting (PGP) endophytes from the *P. hysterophorus* weed growing near the diuron contaminated areas [97]. Among all, endophytes taken under consideration the most efficient isolate namely *Bacillus licheniformis* strain SDS12 degrading around 85.60% of 50 ppm diuron to the benign form by the formation of degradation intermediate 3, 4-dichloroaniline (3, 4-DCA). Similarly, cell free supernatants (CFS) was obtained as a by-product after degradation by strain SDS12 supported the algal growth compared to the pond water. Additionally, it was taken under consideration, that the chlorophyll contents and photosynthetic efficacy of green algae was tested very low in the presence of diuron contaminant water, while no such changes were observed in the SDS12 strain and CFS, respectively. It was suggested that SDS12 bacterial strain can be applied for the degradation and reduction of diuron and its toxicities, especially for the primary producers. More interestingly, the use of this PGP and diuron degrading bacterial isolate from the *P. hysterophorus* could be used in the agriculture fields for the remediation and plants stimulator.

## 9. Conclusions and Future of Prospects

*Parthenium hysterophorus* L. is one of devastating and anxious weed causes severe losses in crop productivity and different diseases in human and animals. The aggressive dominance of this weed is mainly concern with crop production owing its allelochemicals, alarming a serious threads to biodiversity. Different tools were suggested in the last decades for the controlling of this weed, however there is still some possible mechanism like endophytes help and support/tolerate *P. hysterophorus* during adverse conditions, i.e., biotic and abiotic stress. Various endophytic candidates, were taken under consideration that have potential role in the biotic and abiotic stresses condition and thereby improving crops health, growth and development. Endophytes being isolated from *P. hysterophorus* have inoculated in different crops and found to be growth promotive, thereby producing stress tolerating phytohormones like, Indol-3-acetic acid, ACC deaminases, gibberellic acid, cytokinins and abscisic acid. On the same way, these endophytes were also noticed to have strong weedicides and herbicides potential thereby inhibiting the growth and development of some hazardous species. In the current review it was found that weeds like *P. hysterophorus* are rich reservoirs of diverse endophytes which tolerates different crops under stresses conditions. However, these endophytes are not only restricted to be only to scientific research communities of the pharmaceutical and pharmacological industries, but could also be used in the agroforestry’s as bio-fertilizers and bioengineered candidates for sustainable agriculture. 

## Figures and Tables

**Figure 1 microorganisms-10-02217-f001:**
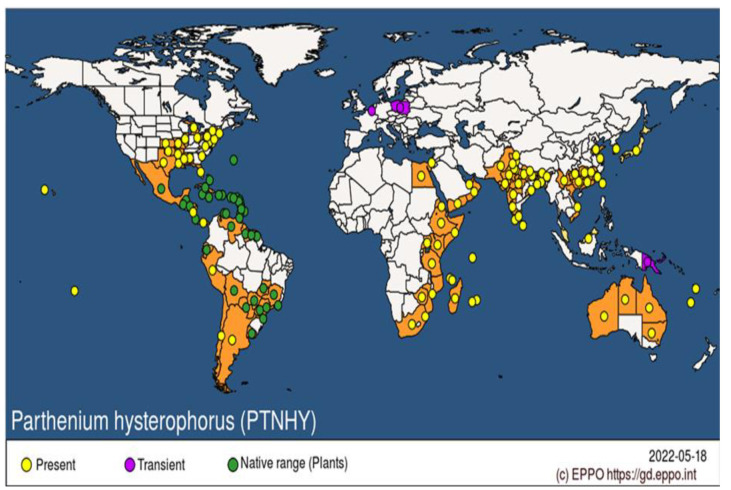
Worldwide distribution of *Parthenium hysterophorus* L. (Source; EPPO https://gd.eppo.int/taxon/PTNHY/distribution; accessed on 29 September 2022).

**Figure 2 microorganisms-10-02217-f002:**
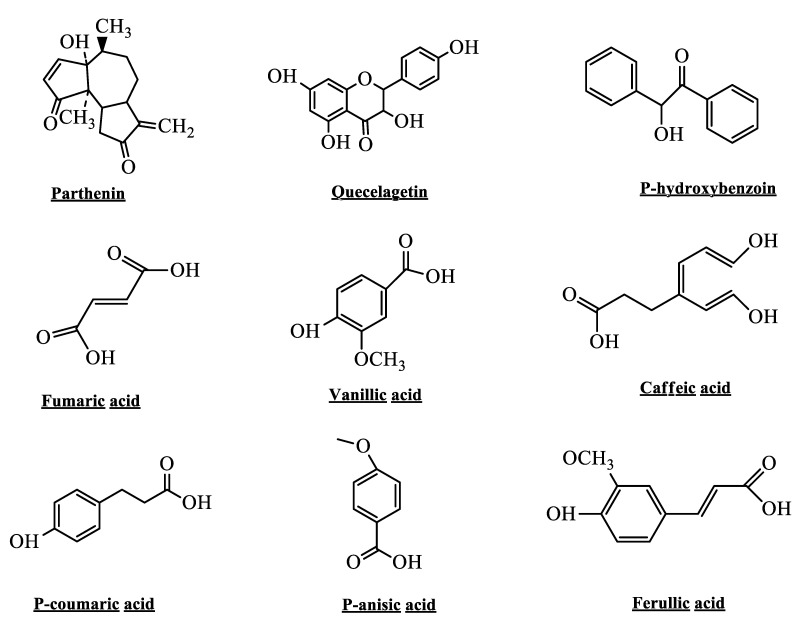
Different phytochemicals isolated from *P. hysterophorus* plants with potential application in the treatment of various health issues.

**Figure 3 microorganisms-10-02217-f003:**
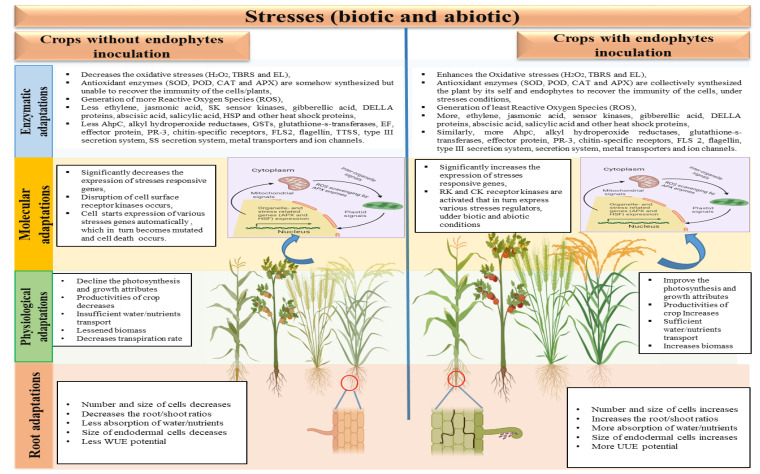
A schematic diagram with comparison of crops with and without inoculation of endophytes under different stressed conditions. Crops exposed to any type of stressed condition reduces roots to shoots length and other physiological and molecular patterns, while the endophytes improve such traits by triggering their antioxidant enzymes and other genetic pathways.

**Figure 4 microorganisms-10-02217-f004:**
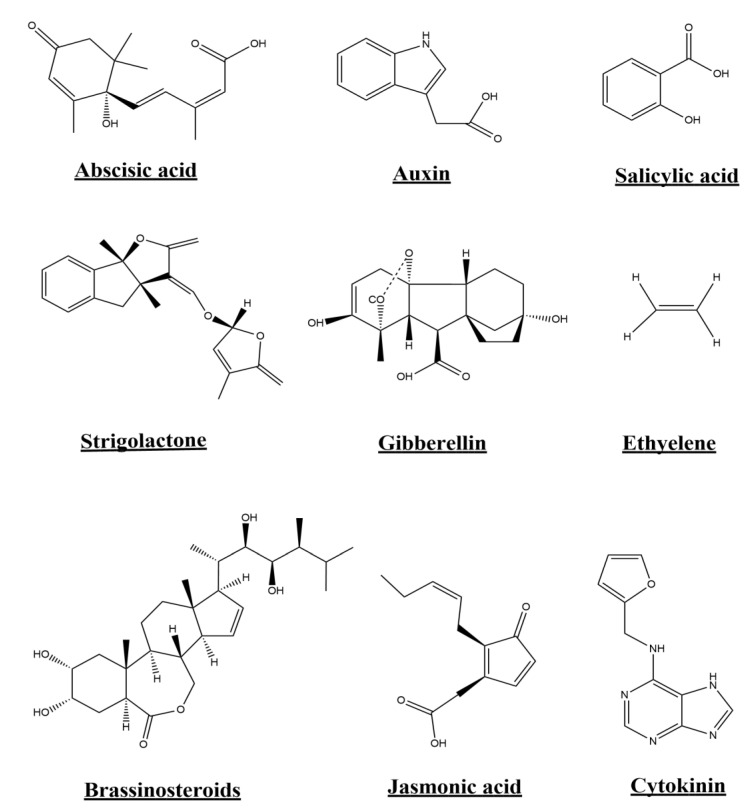
Chemical structures of different classes of stress regulating phytohormes that endophytes could possibly produces, under stresses conditions to enhance host plants growth and deployment.

**Figure 5 microorganisms-10-02217-f005:**
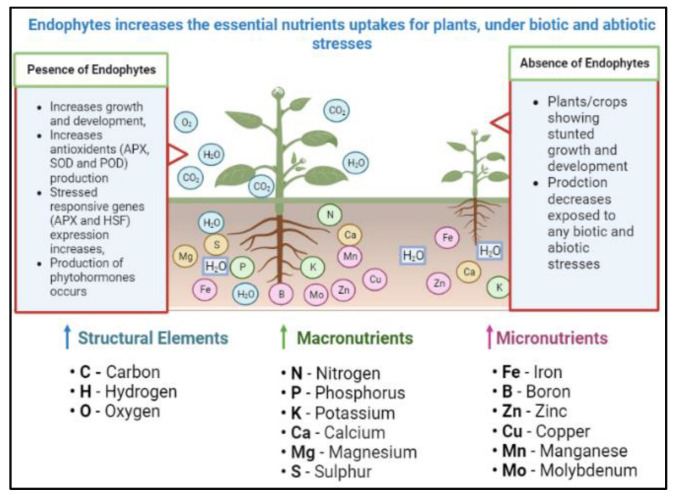
Comparison of different macro and micro heavy metals contaminated soil and with-out plants-endophytes inoculations. Plants inoculated with endophytes have higher growth and comparatively increases the absorption of macro-micro-nutrients than the un-inoculated plants/crops, under heavy metal stressed conditions.

**Table 1 microorganisms-10-02217-t001:** Summary of documented researches on the endophytes from *Parthenium hysterophorus* L. and its role in different biotic and abiotic stresses.

Contents	Endophytes	Target Species	Stresses	Objectives, Results and Conclusion	Reference
Fungal Endophyte	*Curvularia geniculata*	*Cajanus cajan* L.	Phosphorus solubiliztion and its uptake	Objectives; To isolated and characterize DSE (*Curvularia geniculata*) fungi and monitor its P^+^ solubilization, growth promotion and phytohormones production. Results; *C. geniculata* significantly promoted the growth of *Cajanus cajan* plant, while solubilizes P^+^ from different sources and produces significant IAA. Conclusion; This fungus could be used as a bio-inoculate in plant production system.	[21]
	*Alternaria* sp., *Aspergillus* sp., *Drechslera* sp.	*Chenopodium album* L., *Avena fatua* L., *and Convolvulus arvensis* L.	Phytotoxic effect	Objectives; To isolate and identify endophytic fungi and monitored its herbicidal potential of selected weeds. Results; *Alternaria* sp., *Aspergillus* sp., *Drechslera* sp. Were isolated from *P. hysterophorus* roots, among all *Alternaria* showed significant phytotoxic effect, followed by *Drechslera* and *Aspergillus*, respectively. All strains significantly reduces the germination, growth and physiological parameters of the target weeds. Conclusion; The isolated fungal strains could be effective, co-friendly and low-cost biological control of aggressive weeds.	[91]
	*Mucor* sp. *MHR-7*	*Brassica comprestris* L.	Heavy metal stress	Objectives; To isolate endophytic fungi from *P. hysterophorus* and check its heavy metal tolerance. Results; A total of 27 endophytic fungal strains were isolated, among all, *Mucor sp*. MHR-7 was found very effective against Cr, Mn, Co, Zn, and Cu and tolerate upto 900 µg/mL and 60–80% in the 300 µg/mL in broth culture. The fungus produces IAA, ACC deaminase and phosphate solubilization. Conclusion; MHR-7 is an excellent candidate to be use as a bio fertilizer.	[92]
	*Bacillus paramycoides* Ph-04	*Triticum aestivum* L.	Drought stress	Objectives; To isolate effect endophytes from *P. hysterophorus* and check its growth promoting ability in wheat, under drought stress. Results; One bacterial isolate Ph-04 was noted that increases germination, root and shoot growth, under drought condition. Ph-04 inoculated plants showed less membrane integrity and plant damages, produces more APX, SOD, POD and reduces H_2_O_2_ level. Conclusion; *P. hysterophorus* endophytes could be inoculated in crops plants to improve its climatic resilience traits.	[93]
	*Phoma herbarum FGCC#54*	*Lantana camara* L., *Hyptis suavelens* L., *Parthenium hysterophorus* L. and *Sida acuta* Burm. F.	Phyto-toxcity	Objectives; To explore metabolites from the *phoma* sp. Being isolated from parthenium and check its herbicidial activities. Results; Crude extract showed significant phytotoxic potential against the target weeds species. Different compounds, i.e., anhydropseudo-phlegmcin-9, 10-quinone-3-amino-8-O methyl ether ‘anhydropseudophlegmacin-9,10-quinone-3-amino-8-O methyl ether isolated and identified. Conclusion; Different pigments isolated possess significant phytotoxic effect and could be used as a bio herbicides.	[94]
Bacterial Endophyte	*Streptomyces rochei*	Nosocomial pathogens	Anti-microbial activities	Objectives; To isolate endophytic bacteria and check its metabolites production, anti-microbial activities and Polyketide synthase production. Results; A total of 42 bacterial strains were isolated, among all, 12 possess significant anti-microbial activities against *B. subtilis*, *S. aureus*, *E. coli*, *K. pneumonia*, *MRSA*, *C. tropicalis* and *C. vulgari*. All endophytes produces significant phytochemicals peaks *via*; Thin Layer Chromatography and HPLC peaks. Additionally, the molecular screaming revealed the presence of PKS-I gene on the PCR amplification. Conclusion; *P. hysterophorus* being an excellent reservoir of endophytes which produces a number of secondary metabolites which possess significant anti-microbial potential, while the presence of PKS-I gene in such endophytes revealed an unexplored anti-microbial agents.	[95,96]
	*Streptomyces litmocidini*	Nosocomial pathogens	Anti-microbial activities		
	*Streptomyces rochei*	Nosocomial pathogens	Anti-microbial activities		
	*Streptomyces rochei*	Nosocomial pathogens	Anti-microbial activities		
	*Streptomyces enissocaesili*	Nosocomial pathogens	Anti-microbial activities		
	*Streptomyces djakartensis*	Nosocomial pathogens	Anti-microbial activities		
	*Streptomyces olivaceus*	Nosocomial pathogens	Anti-microbial activities		
	*Streptomyces* spp.	Nosocomial pathogens	Anti-microbial activities		
	*Streptomyces plicatus*	Nosocomial pathogens	Anti-microbial activities		
	*Streptomyces geysiriensis*	Nosocomial pathogens	Anti-microbial activities		
	*Streptomyces* spp.	Nosocomial pathogens	Anti-microbial activities		
	*Streptomyces vinaceusdrappus*	Nosocomial pathogens	Anti-microbial activities		
	*Bacillus licheniformis* strain SDS12	Green algae	Diuron toxicity	Objectives; To isolate effective phenylurea herbicides endophytic candidate from *P. hysterophorus* and check its growing potential at diuron degrading sites. Results; *Bacillus licheniformis* SDS12 was found effective improving the growth of green algae *via*; formation of formation of degradation intermediate 3, 4-dichloroanaline (3, 4-DCA). Besides, SDS12 significantly inhibiting the chlorophyll contents and photosynthetic efficiency of the green algae. Conclusion; SDS12 could be used as PGR and diuron degrading candidate in the agriculture sectors.	[97]

**Table 2 microorganisms-10-02217-t002:** MIC (minimal inhibitory concentration) for monitoring the antifungal, antibacterial and antialgal potential of the endophytic actinomycetes isolated from the root and shoot of *Parthenium hysterophorus* L. [95].

Genera	Isolates	*B. subtilis*	*S. aureus* ATCC25923	*E. coli* ATCC 25922	*K. pneumonia* ATCC 706003	*MRSA*	*C. tropicalis*	*C. vulgaris*
*Streptomyces* spp.	RT-6	8	16	16	0	32	8	0
	RT-13	4	8	8	16	8	8	8
	RT-18	8	16	16	0	32	16	0
	RT-36	8	16	16	0	32	32	0
	RT-46	0	16	16	0	16	32	0
	RT-49	16	0	0	0	0	0	0
	RT-50	8	8	16	0	8	8	0
	RT-53	8	16	16	32	16	8	0
	RT-56	8	16	16	32	32	8	0
	RT-57	0	16	16	32	32	32	32
	RT-59	8	16	16	–	32	32	32
	RT-60	8	16	16	32	32	16	16
	RT-63	16	0	0	0	0	0	0
	RT-64	16	0	0	0	0	32	0
	RT-67	8	16	16	32	32	16	16

Where, ‘–’ means no/zero values.

## Data Availability

Not applicable.

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
