# Peer review of "Parthenium hysterophorus’s Endophytes: The Second Layer of Defense against Biotic and Abiotic Stresses"

_microorganisms, 2022, doi:10.3390/microorganisms10112217_

Round 1
Reviewer 1 Report
The manuscript titled “Parthenium hysterophorus’s endophytes: the second layer of defense against biotic and abiotic stresses” reports an interesting work discussing the distribution, advantages and disadvantages of P. hysterophorus, and mainly focuses on the diversity of endophytes isolated from P. hysterophorus and their role in stresses mitigation of other important crops. Various endophytes from P. hysterophorus have been listed their roles in crops, under biotic and abiotic stressed conditions. This is a well-written article and I anticipate that the manuscript should be of great interest to the researchers working on endophytic flora of P. hysterophorus. I include my comments below, most of them are suggestion to improve the overall quality for publication. I considered the manuscript suitable for publication subject to the following improvements.
Specific Comments
In the abstract section: Would be better if, authors merge the two lines that are repeated in the abstract, as follow; 11-13 (The present article reports the distribution, advantages and disadvantages of P. hysterophorus, and mainly focus on the diversity of endophytes isolated from P. hysterophorus and their role in stresses mitigation of other important crops) and line 22-24 (. The aim of this review article is to reveal the potential role of fungal and bacterial endophytes isolated from P. hysterophorus and their capability to mitigate biotic and abiotic stress conditions in crop).
In introduction section, add references to the statement “The invasion of this plant caused the significant supplementation of native/crop plants; increase in the health issues of animal and human beings”.
Restructure the statement “It dispersed in a very short period of time, across the globe (Figure 1)”.
The statement must be revised and references should be added “According to Weyl et al., P. hysterophorus has widespread native range in the USA, Mexico, West Indies and Central America to South America, spreading as far as Argentina and Chile”.
Line 38-41 Try to avoid repetition (animal and human) and shorten if possible.
This statement must be added and elaborated in the said section “Moreover, we have also discussed the phytochemical profile and application of the endophytes in engineering plant microbiome for sustainable agriculture.”
The numbering of each section must be revised also add page and line number to the revised manuscript.
Revise the paragraph and remove the word suffocating “It was found that some weeds like Salvinia molesta, Pistia stratiotes and Eichhornia crasspes causing huge disturbance in the aquatic creatures by suffocating their life cycles [42]. However, P. hysterophorus’ leaf dry extract significantly inhibiting the growth and development of such weeds by wilting and desiccation [43]. Interestingly, it was also found that with the increase in concentration of P. hysterophorus extract the seed’s germination of lovegras also significantly decreased”.
Revise the sentence “These endophytes are noted to produces a number of hormones, enzymes, phytochemicals and iron carriers that directly and indirectly improving the growth and development of plants”.
Figure 3. Must be of high quality and remove extra information and P. hysterophorus, if possible.
Revise the caption of figure 5.
Revise the caption of Figure 4. Different heavy metals contaminated soil with and with-out plants-endophytes inoculations. Plants inoculated with endophytes have comparatively, increases the absorption of macro-micro-nutrients than the un-inoculated plants/crops, under heavy metal stressed condition.
Use the same pattern for all ‘Copper (Cu2), Manganese (Mn), Zinc (Zn2+), chromium (Cr6+) and cobalt (Co2+)’.
In paragraph (from line 200-209) the authors explained very well the phenomena of cell response under oxidative stresses, however, the role of endophytes is elusive. The author needs to shorten the paragraph and focus on endophytes role under such conditions.
Confusing ‘It was found that strain can remove approximately 60-87% of the HMs from the broth culture when applied 300 µgmL-1 of such metals’ (Line 296-297).
Add some latest references.
English language should be edited.
Author Response
Author’s Responses to Reviewer’s Comments
Comments and Suggestions for Authors
Reviewer’s Comments: The manuscript titled “Parthenium hysterophorus’s endophytes: the second layer of defense against biotic and abiotic stresses” reports an interesting work discussing the distribution, advantages and disadvantages of P. hysterophorus, and mainly focuses on the diversity of endophytes isolated from P. hysterophorus and their role in stresses mitigation of other important crops. Various endophytes from P. hysterophorus have been listed their roles in crops, under biotic and abiotic stressed conditions. This is a well-written article and I anticipate that the manuscript should be of great interest to the researchers working on endophytic flora of P. hysterophorus. I include my comments below, most of them are suggestion to improve the overall quality for publication. I considered the manuscript suitable for publication subject to the following improvements.
Author’s Response: We are thankful to you for your positive response. All the suggested changes were incorporated and the specific comments were addressed, accordingly.
Specific Comments
Reviewer’s Comment: In the abstract section: Would be better if, authors merge the two lines that are repeated in the abstract, as follow; 11-13 (The present article reports the distribution, advantages and disadvantages of P. hysterophorus, and mainly focus on the diversity of endophytes isolated from P. hysterophorus and their role in stresses mitigation of other important crops) and line 22-24 (. The aim of this review article is to reveal the potential role of fungal and bacterial endophytes isolated from P. hysterophorus and their capability to mitigate biotic and abiotic stress conditions in crop).
Author’s Response: These lines were merge properly. Thank you
Reviewer’s Comment: In introduction section, add references to the statement “The invasion of this plant caused the significant supplementation of native/crop plants; increase in the health issues of animal and human beings”.
Author’s Response: The statement was cited accordingly. Thank you.
Reviewer’s Comment: Restructure the statement “It dispersed in a very short period of time, across the globe (Figure 1)”.
Author’s Response: The suggested line was rephrased properly. Thank you.
Reviewer’s Comment: The statement must be revised and references should be added “According to Weyl et al., P. hysterophorus has widespread native range in the USA, Mexico, West Indies and Central America to South America, spreading as far as Argentina and Chile”.
Author’s Response: The statement was properly restructured and cited as well. Thank you.
Reviewer’s Comment: Line 38-41 Try to avoid repetition (animal and human) and shorten if possible.
Author’s Response: The repeated information were removed. Thank you.
Reviewer’s Comment: This statement must be added and elaborated in the said section “Moreover, we have also discussed the phytochemical profile and application of the endophytes in engineering plant microbiome for sustainable agriculture.”
Author’s Response: The statement was elaborated accordingly. Thank you.
Reviewer’s Comment: The numbering of each section must be revised also add page and line number to the revised manuscript.
Author’s Response: The said suggestions were added accordingly. Thank you.
Reviewer’s Comment: Revise the paragraph and remove the word suffocating “It was found that some weeds like Salvinia molesta, Pistia stratiotes and Eichhornia crasspes causing huge disturbance in the aquatic creatures by suffocating their life cycles [42]. However, P. hysterophorus’ leaf dry extract significantly inhibiting the growth and development of such weeds by wilting and desiccation [43]. Interestingly, it was also found that with the increase in concentration of P. hysterophorus extract the seed’s germination of lovegras also significantly decreased”.
Author’s Response: The paragraph was revised accordingly and removed the word ‘suffocating’. Thank you
Reviewer’s Comment: Revise the sentence “These endophytes are noted to produces a number of hormones, enzymes, phytochemicals and iron carriers that directly and indirectly improving the growth and development of plants”.
Author’s Response: The said sentence was revised and rephrased, accordingly. Thank you.
Reviewer’s Comment: Figure 3. Must be of high quality and remove extra information and P. hysterophorus, if possible.
Author’s Response: The authors had done their best for improving the quality of the Figure 3. Thank you.
Reviewer’s Comment: Revise the caption of figure 5.
Author’s Response: The said caption was revised, accordingly. Thank you.
Reviewer’s Comment: Revise the caption of Figure 4. Different heavy metals contaminated soil with and with-out plants-endophytes inoculations. Plants inoculated with endophytes have comparatively, increases the absorption of macro-micro-nutrients than the un-inoculated plants/crops, under heavy metal stressed condition.
Author’s Response: The caption was properly revised. Thank you.
Reviewer’s Comment: Use the same pattern for all ‘Copper (Cu2), Manganese (Mn), Zinc (Zn2+), chromium (Cr6+) and cobalt (Co2+)’.
Author’s Response: The said suggestions were addressed properly. Thank you.
Reviewer’s Comment: In paragraph (from line 200-209) the authors explained very well the phenomena of cell response under oxidative stresses, however, the role of endophytes is elusive. The author needs to shorten the paragraph and focus on endophytes role under such conditions.
Author’s Response: The information were properly revised. Thank you.
Reviewer’s Comment: Confusing ‘It was found that strain can remove approximately 60-87% of the HMs from the broth culture when applied 300 µgmL-1 of such metals’ (Line 296-297).
Add some latest references.
Author’s Response: We have revised the sentence properly. Thank you
Reviewer’s Comment: English language should be edited.
Author’s Response: the English language was meticulously improved in the revised manuscript. Thank you
Reviewer 2 Report
Khan et al. wrote a review paper describing the role of endophytes from Parthenium hysterophorus in its adaptability to biotic and abiotic stressors. The paper in its present form is unacceptable for publication. Thus, I encourage the authors to revise this review following the comments stipulated below:
1. Kindly indicate in the introduction the novelty of this review to increase its merit for citations. Is this the first review on Parthenium hysterophorus’ endophytes? If so, better indicate it in the manuscript.
2. Kindly write down the main objectives of this review in the introduction.
3. There are many grammatical errors in the text. I encourage the writers to have this review checked by a professional English editor.
4. The flow of this review is confusing. The title clearly states that this review is focused mainly on the endophytes from Parthenium hysterophorus. However, pages 1-9 were devoted talking all about the weed and not the microbiota within. At this point, this review is rather confusing instead of enlightening.
5. Apart from culture-dependent studies that aimed to identify these endophytes, are there any culture-independent methods (e.g. metagenomics) used to identify the diversity of microflora in P. hysterophorus? Kindly cite those studies.
6. For future prospect, what innovative molecular techniques can you suggest to further study the important role of these endophytes in the adaptability of P. hysterophorus to several biotic and abiotic factors. For example, there have been studies showing that endophyte and host interaction could induce expression of certain biosynthetic genes or pathways to ensure their survival under stress.
7. Apart from future prospect, authors should also include a paragraph highlighting the research gaps / limitations in the study of P. hysterophorus’ endophytes.
8. Kindly cite the following papers highlighting the importance of endophytes in the production of secondary metabolites that have protective roles to host organisms:
Doi: 10.1016/B978-0-12-819541-3.00004-9
Doi: 10.5943/mycosphere/8/1/10
Doi: 10.3390/jof7070572
Doi: 10.2478/botcro-2018-0016
Doi: 10.5943/sif/5/1/14
Author Response
Comments and Suggestions for Authors
Khan et al. wrote a review paper describing the role of endophytes from Parthenium hysterophorus in its adaptability to biotic and abiotic stressors. The paper in its present form is unacceptable for publication. Thus, I encourage the authors to revise this review following the comments stipulated below:
- Kindly indicate in the introduction the novelty of this review to increase its merit for citations. Is this the first review on Parthenium hysterophorus’ endophytes? If so, better indicate it in the manuscript.
Author’s response: We are thankful to you for such a meticulous point. The novelty and related information of the current review are added in the introduction section. Thank you.
- Kindly write down the main objectives of this review in the introduction.
Author’s response: The objectives are well written at the end of introduction. Thank you.
- There are many grammatical errors in the text. I encourage the writers to have this review checked by a professional English editor.
Author’s response: The manuscript was revised and checked by one of our colleague for English editing. Thank you.
- The flow of this review is confusing. The title clearly states that this review is focused mainly on the endophytes from Parthenium hysterophorus. However, pages 1-9 were devoted talking all about the weed and not the microbiota within. At this point, this review is rather confusing instead of enlightening.
Author’s response: Yes, we had tried to answer, why we choose Parthenium hysterophorus? For this, we mention some information to reflect the importance of P. hysterophorus. By discussing different aspects like, positive and negative aspects. In the recent, years a number of ecological and molecular strategies/mechanism were proposed, however, we reported the importance of its endophytic flora and their role under biotic and abiotic stresses. Thank you.
- Apart from culture-dependent studies that aimed to identify these endophytes, are there any culture-independent methods (e.g. metagenomics) used to identify the diversity of microflora in P. hysterophorus? Kindly cite those studies.
Author’s response: We are thankful to you for your kind suggestion. However, the metagenomics identification of the microflora in P. hysterophorus is not included in our objectives and we haven’t added in this manuscript. We have planned another article in near future subject to your kind suggestions.
- For future prospect, what innovative molecular techniques can you suggest to further study the important role of these endophytes in the adaptability of P. hysterophorusto several biotic and abiotic factors. For example, there have been studies showing that endophyte and host interaction could induce expression of certain biosynthetic genes or pathways to ensure their survival under stress.
Author’s response: With reference to our previous comment the molecular techniques and Omics (Genomics, Transcriptomic, Proteomics, and metabolomics) based study will be the focus of our upcoming article. Where we will be able to reveal that endophyte and host interaction could induce expression of certain biosynthetic genes or pathways to ensure their survival under stress. Thank you.
- Apart from future prospect, authors should also include a paragraph highlighting the research gaps / limitations in the study of P. hysterophorus’ endophytes.
Author’s response: The research gaps and limitations in the study of P. hysterophorus’ endophytes have been added in the revised manuscript. Thank you.
- Kindly cite the following papers highlighting the importance of endophytes in the production of secondary metabolites that have protective roles to host organisms: Doi: 10.1016/B978-0-12-819541-3.00004-9; Doi: 10.5943/mycosphere/8/1/10; Doi: 10.3390/jof7070572; Doi: 10.2478/botcro-2018-0016; Doi: 10.5943/sif/5/1/14
Author’s response: All these articles were found interesting and cited, accordingly. Thank you.
Reviewer 3 Report
MS # microorganisms-2028545 comments
Abstract
1. (This weed utilizes most capable endophytic flora as an additional line), the author should add some short possible mechanism of plants its self for example allechemicals etc.
2. The author claiming the isolated endophytes from Parthenium hysterophorus role in different stresses and then directly saying about enzymes responsible for the beneficial effects. Can the authors add some growth attributes/aspect that improved by such endophytes (The beneficial role of the endophytes).
3. Just ‘indol-3-acetic acid and ACC deaminase’ were found be the growth responsible enzymes, if/yes please justify.
Introduction
Over all introduction is good, however, authors need to rearrange it accordingly and there are a lot of repeated information.
1. The first two paragraphs are repetitive, try to merge or/shorten. Would be better if the second paragraphs come first (starting from broaden to specific).
2. ‘Parthenium hysterophorus L. is one of the most devastating weed in the tropical and sub-tropical regions of the world, belongs to the family Asteraceae, native to tropical America. How this is possible for a plant to be weed in tropical regions and native to tropical regions at the same time, please refine this claim.
3. In third and fourth para the information are repeated again, try to avoid.
4. Again the information concerning the seeds production repeated, try to avoid and refine.
3. Positive aspects of Parthenium. hysterophorus L.
1. Parthenium. hysterophorus, remove ‘,’ from the plant name.
2. Rephrase the statement (Additionally, due to rich phytochemicals like, phenolic acids, sesquiterpene, lactones like parthenin and coronopilin), also what does ‘rich’ means?.
3. Try to use the same phrase for the plant, replace Parthenium with P. hysterophorus, as repeated in the whole manuscript.
4. Negative aspects of Parthenium hysterophorus L
1. In paragraph no 1, rephrase the sentence.
2. The stating slaying are the same (Due to the..) for the 2nd paragraph.
3. In the fourth para it would be better if these information comes in the section ‘Impact on live stock’
4. Also, move the above section to ‘Impact on soil microflora’.
5. Complete the sentence ‘and other related compounds in fractions’.
6. Try to improve the quality of Figure 2.
5. Endophytic flora and their diverse activities, under stressful conditions
1. Revise and elaborate the statement ‘Endophytes are the class of symbiotic microbes, especially the bacteria and fungi actively participating in the maintenance in the normal physio-chemical traits of plants’. Do author’s mean to say endophytes participates in plants function even in normal conditions?, if yes please elaborates?
2. what does mean ‘won’?
3. Figure-3 needs to improve the overall quality and focus on few corrections; i.e. boletes are not the same, remove the background bundies, especially, in the root section (also check over all), check the spells.
6. Endophytic flora of Parthenium hysterophorus L.
1. It is suggested to elaborate the stamen ‘A number of different fungal and bacterial endophytes being reported from P. hysterophorus and evaluated, under biotic and abiotic stressed conditions on different crops’. I think, this information will be suited if comes after ‘potential against various biotic and abiotic stresses’.
2. Shorten the sentence ‘Interestingly, more recently endophytes were found to the abilities to enhance the growth and development of crops via; germination rate, fresh biomass, chlorophyll contents, desolation of essential metals (Fe, Ca. Mg, P, S and K) and other importance minerals in the heavy meatal (Cd, Mo, Zn, Cu, Hg, Mn, Br and Mg) contaminant soils’.
3. Improve the quality of the figure 4.
4. Cite the references in ‘A number of studies were taken under consideration to document endophytes and check its secondary metabolites production and their potential toward different stressed conditions.’
5. Table 1, Fungal and bacterial endophytes and also citations needs center amendments.
6. Table 1, some plants names needs to be completed i.e. authority names are missing e.g Chenopodium album.
7. Check spells of ‘Dadmium’,
8. What authors means ‘a number of bacterial’, please specify the actual number.
9. Bacillus paramycoides needs to italic.
10. What authors means from ‘cytokines’ or cytokinin???
Author Response
Comments and Suggestions for Authors
Abstract
- (This weed utilizes most capable endophytic flora as an additional line), the author should add some short possible mechanism of plants its self for example allechemicals etc.
Author’s response: We are thankful to you for your positive response. We have added some terminology regarding the weeds capabilities that helps in the managing of different stresses.
- The author claiming the isolated endophytes from Parthenium hysterophorus role in different stresses and then directly saying about enzymes responsible for the beneficial effects. Can the authors add some growth attributes/aspect that improved by such endophytes (The beneficial role of the endophytes).
Author’s response: Different attributes concerning the growth traits of host/crops are mentioned, accordingly. Thank you.
- Just ‘indol-3-acetic acid and ACC deaminase’ were found be the growth responsible enzymes, if/yes please justify.
Author’s response: Thank you for raising such an important point. We have elaborated with examples, accordingly.
Introduction
Over all introduction is good, however, authors need to rearrange it accordingly and there are a lot of repeated information.
- The first two paragraphs are repetitive, try to merge or/shorten. Would be better if the second paragraphs come first (starting from broaden to specific).
Author’s response: Yes, We are completely agree on this point and hence removed the first paragraphs and updated the missing information in the next paragraphs, accordingly. Thank you.
- ‘Parthenium hysterophorus is one of the most devastating weed in the tropical and sub-tropical regions of the world, belongs to the family Asteraceae, native to tropical America. How this is possible for a plant to be weed in tropical regions and native to tropical regions at the same time, please refine this claim.
Author’s response: As these lines/information were coming in the first paragraphs (which is already been removed), so rectified accordingly. Thank you
- In third and fourth para the information are repeated again, try to avoid.
Author’s response: The repeated information were removed, accordingly. Thank you.
- Again the information concerning the seeds production repeated, try to avoid and refine.
Author’s response: Whole introduction section was revised and repeated information were removed, accordingly. Thank you.
- Positive aspects of Parthenium. hysterophorusL.
- Parthenium. hysterophorus, remove ‘,’ from the plant name.
- Rephrase the statement (Additionally, due to rich phytochemicals like, phenolic acids, sesquiterpene, lactones like parthenin and coronopilin), also what does ‘rich’ means?.
- Try to use the same phrase for the plant, replace Partheniumwith P. hysterophorus, as repeated in the whole manuscript.
Author’s response:
- The comma/dot being removed,
- The sentence was rephrased, accordingly,
- The name Parthenium was replace by hysterophorus in whole manuscript. Thank you.
- Negative aspects of Parthenium hysterophorus L
- In paragraph no 1, rephrase the sentence.
- The stating slaying are the same (Due to the..) for the 2ndparagraph.
- In the fourth para it would be better if these information comes in the section ‘Impact on live stock’
- Also, move the above section to ‘Impact on soil microflora’.
- Complete the sentence ‘and other related compounds in fractions’.
- Try to improve the quality of Figure 2.
Author’s response;
- Thank you for your positive comments. Rephrased, accordingly.
- Removed the starting slaying and also rephrased the sentences.
- The said line/information is cut and pasted in the said place/section, accordingly.
- The whole sentence was refine and made more meaningful.
- The quality of Figure 2 was improved, accordingly.
- Endophytic flora and their diverse activities, under stressful conditions
- Revise and elaborate the statement ‘Endophytes are the class of symbiotic microbes, especially the bacteria and fungi actively participating in the maintenance in the normal physio-chemical traits of plants’. Do author’s mean to say endophytes participates in plants function even in normal conditions?, if yes please elaborates?
- what does mean ‘won’?
- Figure-3 needs to improve the overall quality and focus on few corrections; i.e. boletes are not the same, remove the background bundies, especially, in the root section (also check over all), check the spells.
Author’s response:
- The sentence was properly resided and rephrased, accordingly. Due to the lack of proper information to claim whether endophytes had some role in plants or not under normal conditions. However, these endophytes produce different phytohormones and enzymes but also have a key role in modulating different traits of their host, under stressed conditions,
- The won was replaced by own,
- We tried our best to improve all figures including figure 3. Thank you.
- Endophytic flora of Partheniumhysterophorus L.
- It is suggested to elaborate the stamen ‘A number of different fungal and bacterial endophytes being reported from P. hysterophorusand evaluated, under biotic and abiotic stressed conditions on different crops’. I think, this information will be suited if comes after ‘potential against various biotic and abiotic stresses’.
- Shorten the sentence ‘Interestingly, more recently endophytes were found to the abilities to enhance the growth and development of crops via; germination rate, fresh biomass, chlorophyll contents, desolation of essential metals (Fe, Ca. Mg, P, S and K) and other importance minerals in the heavy meatal (Cd, Mo, Zn, Cu, Hg, Mn, Br and Mg) contaminant soils’.
- Improve the quality of the figure 4.
- Cite the references in ‘A number of studies were taken under consideration to document endophytes and check its secondary metabolites production and their potential toward different stressed conditions.’
- Table 1, Fungal and bacterial endophytes and also citations needs center amendments.
- Table 1, some plants names needs to be completed i.e. authority names are missing e.g Chenopodium album.
- Check spells of ‘Dadmium’,
- What authors means ‘a number of bacterial’, please specify the actual number.
- Bacillus paramycoides needs to italic.
- What authors means from ‘cytokines’ or cytokinin???
Author’s response;
- Yes, the said information was removed and incorporated in the section potential against various biotic and abiotic stresses.
- This line was well written in various independent smaller lines,
- Improved,
- Since different groups have reported these endophytes from Parthenium which were comprehensively listed in the Supplementary file and as well as in Table 1. Here the authors have cited the supplementary line and table, where proper citations are mentioned,
- The citations were properly centered amendment,
- ‘The Plant list and literature’ were properly checked for the listed plants and their authority names were assigned to the mentioned plants in the table,
- Corrected by Cadmium
- A total of 21 strains were isolated and properly mentioned in the said place,
- Italic it accordingly,
- Replaced/corrected with Cytokinins
All suggested information was incorporated accordingly, thank you.
Round 2
Reviewer 2 Report
The paper has been extensively revised and is now acceptable for publication.